# Effect of Isoflavone on Muscle Atrophy in Ovariectomized Mice

**DOI:** 10.3390/nu16193295

**Published:** 2024-09-28

**Authors:** Sayaka Kawai, Takuro Okamura, Chihiro Munekawa, Yuka Hasegawa, Ayaka Kobayashi, Hanako Nakajima, Saori Majima, Naoko Nakanishi, Ryoichi Sasano, Masahide Hamaguchi, Michiaki Fukui

**Affiliations:** 1Department of Endocrinology and Metabolism, Graduate School of Medical Science, Kyoto Prefectural University of Medicine, Kyoto 602-8566, Japan; sayaka25@koto.kpu-m.ac.jp (S.K.);; 2AiSTI SCIENCE Co., Ltd., Wakayama 640-0033, Japan

**Keywords:** estrogen, gut microbiota, isoflavone, menopause, sarcopenia

## Abstract

Background: Sarcopenia, characterized by muscle mass decline due to aging or other causes, is exacerbated by decreased estrogen levels after menopause in women. Isoflavones, a class of flavonoids acting on estrogen receptors, may have beneficial effects on metabolic disorders. We examined these effects in ovariectomized mice fed a high-fat, high-sucrose diet (HFHSD). Methods: At 7 weeks old, female C57BL6/J mice (18–20 g, *n* = 12) underwent bilateral ovariectomy (OVX), and were then fed a high-fat, high-sucrose diet starting at 8 weeks of age. Half of the mice received isoflavone water (0.1%). Metabolic analyses, including glucose and insulin tolerance tests, were conducted. Muscle analysis involved grip strength assays, next-generation sequencing, quantitative RT–PCR, and western blotting of skeletal muscle after euthanizing the mice at 14 weeks old. Additionally, 16S rRNA gene sequence analysis of the gut microbiota was performed. Results: The results demonstrated that isoflavone administration did not affect body weight, glucose tolerance, or lipid metabolism. In contrast, isoflavone-treated mice had higher grip strength. Gene expression analysis of the soleus muscle revealed decreased Trim63 expression, and western blotting showed inactivation of muscle-specific RING finger protein 1 in isoflavone-treated mice. Gut microbiota analysis indicated higher Bacteroidetes and lower Firmicutes abundance in the isoflavone group, along with increased microbiota diversity. Gene sets related to TNF-α signaling via NF-κB and unfolded protein response were negatively associated with isoflavones. Conclusions: Isoflavone intake alters gut microbiota and increases muscle strength, suggesting a potential role in improving sarcopenia in menopausal women.

## 1. Introduction

Sarcopenia refers to a condition in which muscle mass decreases due to aging or other factors. Its prevalence is estimated to be 10% of the world’s population in both men and women [1]. Diabetes is a well-established risk factor for sarcopenia. Diabetes is also known to accelerate age-related decline in skeletal muscle mass and function [2]. The incidence of type 2 diabetes is increasing, and as the population ages, advancing its treatment and prevention of complications is imperative.

Estrogen is a sex steroid hormone. Menopause is caused by age-related changes in estrogen levels [3], and decreased estrogen levels are recognized as an exacerbating factor for sarcopenia [4]. Furthermore, it is known that decreased estrogen levels tend to increase the occurrence of a variety of diseases, including cardiovascular disease, dementia, metabolic syndrome, osteoporosis, sarcopenia, and frailty [4,5,6,7]. For the prevention of sarcopenia, dietary and exercise interventions are generally recommended [8]. Furthermore, some studies have reported that hormone replacement therapy (HRT) with estrogen is effective in preventing a decrease in the amount and strength of skeletal muscle in postmenopausal women [9,10,11]. Recently, significant progress has been made in the development of dietary supplements for the prevention of age-related sarcopenia.

Isoflavones are flavonoids that are abundant in soy. The structure of isoflavones is similar to that of estrogen. Isoflavones exert various physiological effects by acting on the estrogen receptors and are reported to have beneficial effects on glucose tolerance and hypertension [12,13,14].

There is some evidence to suggest a link between isoflavones and the prevention of muscle weakness [15]. For example, it has been reported that daidzein, a type of soy isoflavone, reduces lipid accumulation in muscle cells [16]. In addition, isoflavone treatment markedly inhibited TNF-α-induced MuRF1 promoter activity and reduced myotube atrophy in C2C12 myotube cells [17].

The bioavailability of isoflavones can be altered by the fermentation of isoflavones in the gut microbiota and their metabolism into more biologically active forms [18]. Furthermore, numerous reports have documented the correlation between sarcopenia and gut microbiota [19,20,21]. So, concerning the relationship between isoflavone administration and the prevention of muscle wasting, it is imperative to consider both the direct nutritional impact of isoflavone consumption and the metabolic effects mediated by the gut microbiota.

We hypothesized that isoflavones may prevent sarcopenia associated with female menopause. In this study, we conducted various experiments to perform multi-omics research on isoflavones in sarcopenia, with a particular focus on the gut microbiota, the expression of muscle atrophy-related genes, and isoflavone-related metabolites.

## 2. Materials and Methods

### 2.1. Mice and Ovariectomy

All animal experiments were conducted following approval from the Animal Research Committee at Kyoto Prefectural University of Medicine (M2023-78). Female C57BL6/J mice, 6 weeks old, weighing between 18 and 20 g (*n* = 12), were sourced from SHIMIZU Laboratory Supplies Co., Ltd. (Kyoto, Japan). We obtained mice in groups of six at weekly intervals. The mice were maintained in the animal care facility of Kyoto Prefectural University of Medicine under specific pathogen-free conditions, with the temperature controlled at 23 ± 1.5 °C, and exposed to a 12 h light/dark cycle (7 a.m. to 7 p.m.). Mice were housed in cages of W220 × L320 × H135 (mm), with six mice in each cage. During the first week, the mice were acclimatized, and at 7 weeks of age, all mice were ovariectomized by administering an anesthetic combination of 0.3 mg/kg medetomidine, 4.0 mg/kg midazolam, and 5.0 mg/kg butorphanol to produce an estrogen-deficient state.

From 8 weeks of age, the first six OVX mice were fed a high-fat, high-sucrose diet (HFHSD, 459 kcal/100 g, 20% protein, 40% carbohydrate, 40% fat; fat, D12 327, Research Diets, Inc., New Brunswick, NJ, USA) and normal water for 6 weeks (control group *n* = 6). The amount of food was measured twice a week. The remaining six OVX mice were fed the HFHSD and soy isoflavone water (0.1%) (isoflavone group *n* = 6) from 8 weeks of age for 6 weeks. The isoflavone group was given the same amount of food as the control group at the same age. The weight of both groups was measured twice a week. The soy isoflavones have an isoflavone content of 40.74% (Soyaflavone HG; Fuji Foundation for Protein Research), and the composition of isoflavones in Soyaflavone HG consists of various forms of isoflavones measured in grams per kilogram (g/kg). The most abundant isoflavones are malonyl daidzin, which is present at 361.7 g/kg, and daidzin at 239.4 g/kg. Other significant isoflavones include glycitin at 95.9 g/kg, malonyl glycitin at 135.9 g/kg, and genistin at 73.4 g/kg. Less abundant forms, such as acetyl daidzin (10.7 g/kg), acetyl glycitin (9.3 g/kg), and acetyl genistin (2.5 g/kg), are also present. Additionally, trace amounts of the aglycones daidzein (1.8 g/kg), genistein (0.1 g/kg), and glycitein (0.5 g/kg) are found in the composition. Soyaflavone HG was dissolved in water to a concentration of 0.1%. Previous reports have shown that mice fed an HFHSD showed signs of sarcopenia [22], and that consumption of an HFHSD is closely related to worsening insulin resistance and inflammation of muscle tissue [23]. These mice are a model for the elderly with metabolic diseases caused by the effects of modern Westernized diets, and we thought they would be suitable for studying the effects of isoflavones on menopausal women. For this reason, we chose the HFHSD. The HFHSD used in this study contained soybean oil. However, previous reports have shown that the amount of isoflavones contained in soybean oil is very small, so there is no need to be concerned about the amount of isoflavones ingested from the HFHSD [24]. Power analysis was conducted based on the mean and standard deviation of the area under the curve (AUC) of the intraperitoneal glucose tolerance test (IPGTT) performed at 13 weeks of age in the two groups. With a significance level of 0.05 and a power of 80%, it was calculated that a minimum of six biological replicates per group would be required to detect a statistically significant difference. Consequently, the experiment was carried out using six animals per group (*n* = 6). When the mice reached 14 weeks of age, after fasting for 16 h, they were euthanized by administration of a combination anesthetic of 0.3 mg/kg medetomidine, 4.0 mg/kg midazolam, and 5.0 mg/kg butorphanol. All investigators were not blinded to the experimental conditions. There were no mice in any of the experimental groups that were not included in the analysis.

### 2.2. Analytical Procedures and Glucose and Insulin Tolerance Tests

In 13-week-old mice, we performed an intraperitoneal glucose tolerance test (IPGTT; 1 g/kg body weight) and an insulin tolerance test (ITT; 0.5 U/kg body weight) following fasting periods of 16 h and 5 h, respectively. Blood glucose levels were recorded at predetermined intervals using a glucometer (FreeStyle Libre Reader, Abbott Diabetes Care, Witney, UK), which involved collecting blood droplets. The areas under the curve (AUC) for both the IPGTT and ITT results were subsequently analyzed.

### 2.3. Determinations of Biochemical Assays

Immediately after euthanasia, blood was collected by cardiac puncture. Serum was prepared by centrifugation at 5000 rpm for 20 min at 4 °C. The serum concentrations of albumin were measured using the bromocresol green method, those of alanine aminotransferase were measured using the Japan Society of Clinical Chemistry JSCC transferable method, and those of triglycerides, total cholesterol, and nonesterified fatty acids were measured using enzymatic methods. Serum hormone estradiol levels were measured using an enzyme-linked immunosorbent assay (ELISA) assay kit (R&D Systems, Elabscience Biotechnology Co., Ltd., Wuhan, China) according to the manufacturer’s instructions.

### 2.4. Histological Analysis

The plantaris muscle was removed, treated with 10% buffered formalin for fixation, and subsequently embedded in paraffin. Thin muscle sections were then prepared and stained using hematoxylin and eosin (H&E). Images were obtained through a BZ-X710 fluorescence microscope (Keyence), with two random images captured per sample. The cross-sectional area of the myofibers from these images was measured and analyzed using ImageJ software (RRID: SCR_003070, National Institutes of Health, Bethesda, MD, USA).

### 2.5. Grip Strength Assay

To measure grip strength, we used a mouse grip strength meter (model DS2-50N, Imada Manufacturing Co., Ltd., Toyohashi City, Aichi Prefecture, Japan) and measured grip strength at 14 weeks of age. We performed three consecutive measurements at one-minute intervals and standardized grip strength according to body weight.

### 2.6. Next-Generation Sequencing and Quantitative RT–PCR

The soleus muscle from murine models was carefully extracted and immediately frozen in liquid nitrogen. The tissue samples were then homogenized using ice-cold QIAzol Lysis Reagent (Qiagen, Venlo, The Netherlands) at a speed of 4000 rpm for 2 min in a ball mill. Total RNA was subsequently extracted following the manufacturer’s guidelines. Complementary DNA (cDNA) libraries were generated using the TruSeq^®^ Stranded mRNA kit (Qiagen, Carlsbad, CA, USA), and paired-end sequencing was performed with the Illumina NovaSeq6000 platform (*n* = 6). To assess the functional enrichment of highly expressed genes, genes were ranked by their log-fold changes, and gene set enrichment analysis (GSEA) was carried out using previously identified gene markers [25]. A positive normalized enrichment score (NES) indicated gene enrichment, while a negative NES indicated gene depletion. The soleus muscle was selected for this analysis due to its lower fat content, which reduces the risk of contamination compared to the gastrocnemius muscle.

### 2.7. Protein Extracts and Western Blot Incubated with Antibodies

The gastrocnemius muscles from both legs were homogenized and extracted in a radioimmunoprecipitation assay (RIPA) buffer [ATTO, Tokyo; 50 mmol/L Tris (pH 8.0), 150 mmol/L NaCl, 0.5% deoxycholate, 0.1% sodium dodecyl sulfate (SDS), and 1.0% NP-40] supplemented with a protease inhibitor cocktail (BioVision, Milpitas, CA, USA). Protein concentrations were determined using a bovine serum albumin (BSA) protein assay kit (Pierce, Thermo Scientific, Rockford, IL, USA) following the manufacturer’s instructions. Total protein was separated by 12% sodium dodecyl sulfate–polyacrylamide gel electrophoresis (SDS-PAGE), transferred onto membranes, and subjected to western blotting using standard protocols. Protein bands were visualized with ImageQuant LAS 500 (GE Healthcare, Piscataway, NJ, USA). Initially, 10–20 μg of extracted protein was incubated with monoclonal primary anti-MuRF1 antibodies (1:500) diluted in EzBlock Chemi (ATTO, Osaka, Japan) overnight at 4 °C. (MuRF-1, also known as TRIM63, is a RING-type E3 ligase that induces muscle atrophy.) Subsequently, the cells were incubated at room temperature for 60 min with a diluted horseradish peroxidase-conjugated goat anti-mouse IgG secondary antibody. The protein expression levels were assessed by measuring optical density with ImageJ (NIH). All antibodies utilized in this study were sourced from Santa Cruz Biotechnology (Dallas, TX, USA).

### 2.8. Quantification of Isoflavone Concentrations in Serum

Compositional analysis of isoflavones in murine serum was performed using gas chromatography–mass spectrometry (GC/MS) on an Agilent 7890B/7000D system (Agilent Technologies, Santa Clara, CA, USA). We subjected a mixture of 50 µL of serum and 150 µL of distilled water to shaking for a few seconds, followed by addition of 800 μL of acetonitrile. The samples were subsequently agitated at 1000 rpm for 30 min at 37 °C and then centrifuged at 156× *g* for 3 min at ambient temperature. The pH of the mixture was adjusted to 9 with 0.1 mol/L NaOH, after which isoflavone was extracted. The concentrations of isoflavones were determined by GC/MS, employing an online solid-phase extraction (SPE) method. In the SPE-GC system SGI-M100 (AiSTI SCIENCE, Wakayama, Japan), SPE and injection into the GC/MS system were automatically performed after the sample was added to the vial and set on an autosampler tray. Flash-SPE ACXs (AiSTI SCIENCE) were used for solid-phase stratification. To measure the concentrations of isoflavone, 50 µL aliquots of each sample extract were loaded onto the solid phase and washed with a 1:1 mixture of acetonitrile and water. The samples underwent dehydration using acetonitrile, followed by treatment with 4 μL of a 0.5% methoxyamine–pyridine solution. After this step, N-methyl-N-trimethylsilyltrifluoroacetamide was introduced to the solid phase, facilitating both methoxylation and trimethylsilylation during the derivatization process, with subsequent elution performed using hexane. The resulting product was injected via the LVI-S250 programmed temperature vaporizer injector (AiSTI SCIENCE), where it was held at 220 °C for 0.5 min. The temperature was then ramped up at a rate of 50 °C/min to reach 290 °C, which was maintained for an additional 16 min. The samples were placed onto a capillary column (Vf-5 ms; 30 m × 0.25 mm [inner diameter] × 0.25 μm [film thickness]; Agilent Technologies). The initial column temperature was set at 80 °C for 3 min, then increased at a rate of 25 °C/min to 190 °C, followed by a 3 °C/min rise to 220 °C, and finally ramped at 15 °C/min to 310 °C, where it was held for 4.6 min. The injection process was carried out in split mode, with a split ratio of 50:1. The specimens underwent dehydration with acetone, followed by impregnation with a 4 µL solution of N-tert-butyldimethylsilyl-N-methyltrifluoroacetamide (MTBSTFA)-toluene (1:3 ratio), and subsequent elution with hexane following derivatization on the solid phase. The resultant mixture was then introduced using a programmed temperature vaporizer injector, specifically the LVI-S250 model by AiSTI SCIENCE, where the temperature was held at 150 °C for 0.5 min, then gradually increased at a rate of 25 °C per minute until reaching 290 °C, and maintained at this temperature for 16 min. The samples were loaded onto a capillary column, Vf-5 ms (30 m  ×  0.25 mm [inner diameter]  ×  0.25 μm [membrane thickness]; Agilent Technologies). The column temperature was maintained at 120 °C for 3 min, ramped at 40 °C/min to 2800 °C and at 10 °C/min to 320 °C, and then held there for 2 min. The sample was injected in the split mode at a split ratio of 20:1. Isoflavone was detected in the scan mode (*m/z* 70–490).

### 2.9. Gut Microbiota Analysis

Immediately after euthanasia, fecal samples were meticulously obtained from the appendix and subsequently transferred to cryotubes. Immediately after procurement, the samples were cryopreserved in liquid nitrogen and stored until DNA extraction. Microbial DNA was assiduously isolated from the cryopreserved fecal samples using the QIAamp^®^ DNA Stool Mini Kit (Qiagen, Venlo, The Netherlands), adhering rigorously to the manufacturer’s specified protocol.

Whole-genome shotgun sequencing was performed using a HiSeq 2000/2500/4000 system (Illumina, San Diego, CA, USA) at the Bioengineering Lab. Co., Ltd., Sagamihara, Japan. QIIME version 1.9.1 was employed to enhance sequence quality. Barcodes or primers with scores below 75% were meticulously excised from the files. Operational taxonomic units (OTUs) were identified using the UCLUST algorithm with a 97% similarity threshold [26]. Predictions regarding the abundance of Kyoto Encyclopedia of Genes and Genomes (KEGG) orthologs were made using the Phylogenetic Investigation of Communities by Reconstruction of Unobserved States (PICRUSt2) software (version 2.5.2) [27]. The relative abundance of phyla across the cohorts was analyzed with a one-way analysis of variance (ANOVA), followed by a Holm–Šídák multiple-comparison test. Alpha diversity, which reflects diversity within individual samples, was evaluated using the Chao1 [28], Shannon [29], Gini–Simpson [30], and Good’s coverage metrics [31].

The comparative abundance of bacterial genera between groups was analyzed using Linear Discriminant Analysis (LDA) combined with effect size measurements (LEfSe) (http://huttenhower.sph.harvard.edu/lefse/, accessed on 15 May 2023) [32]. LEfSe processed a normalized relative abundance matrix, identified taxa exhibiting significant differences in abundance, and assessed the effect size of each feature using LDA. A *p*-value threshold of 0.05 (Wilcoxon rank-sum test) and an effect size threshold of 2 were established for all biomarkers identified in this study.

### 2.10. Network Analysis and Visualization Methodology

Co-abundant gene groups (CAGs) were identified to elucidate potential groups of bacteria that tended to be abundant or scarce concurrently within the microbial communities. Network analysis and visualization were conducted using Python (Python 3.6+), utilizing Pandas (version 1.5.3) for data manipulation, NetworkX (version 2.8.6) for network analysis, and Matplotlib (version 3.6.0) for visualization. Initially, the correlation data were filtered to retain only pairs with a correlation coefficient ≥ 0.4. Subsequently, within each CAG, bacteria were ranked based on their mean abundance, and the top seven were selected for inclusion in the network visualization. A graph was then constructed, where nodes represented the selected bacteria and edges represented significant correlations between them, with edge width proportional to the correlation coefficient. Node sizes were determined by the log2-transformed mean abundance values of the corresponding bacteria, scaled for visual clarity, and colored distinctly according to their CAG using a predefined color palette (matplotlib’s tab20). The graph employs the Kamada–Kawai layout algorithm for node positioning, which optimizes the distance between all pairs of nodes to create a balanced and readable visualization. The final visualization included a legend indicating the color corresponding to each CAG, and various custom adjustments were applied to the node sizes, edge widths, and colors to enhance the clarity and aesthetics of the figures.

### 2.11. Generation of Circos Plots

Circos plots were utilized to comprehensively describe the complex relationships and patterns in the multidimensional dataset; only pairs with correlation coefficients > 0.4 or < −0.4 in CAG, metabolites in stool, and nutrient absorption transporters that showed predominant expression changes between the two groups were drawn.

### 2.12. Statistical Analysis

Data underwent analysis using GraphPad Prism Version 10 (GraphPad Software, La Jolla, CA, USA). Welch’s *t*-test was applied for comparisons between two groups. The findings were delineated as the means ± standard deviations (SD). *p* values below 0.05 were considered statistically significant. Figures were generated using GraphPad Prism Version 10.

## 3. Results

### 3.1. Administration of Isoflavone Did Not Change Body Weight, Glucose Tolerance, and Lipid Metabolism in OVX Mice

Since the mice were fed in pairs, no discernible differences were noted in the body weights of mice administered HFHSD with isoflavone compared to those given HFHSD without isoflavone (Figure 1a). In fact, there was no significant difference in food intake between the two groups (Figure 1b). No significant variations in blood glucose levels, as assessed by iPGTT and ITT, were detected between the two groups (Figure 1c,d). Similarly, no significant differences were observed in albumin, alanine aminotransferase, triglyceride, total cholesterol, or nonesterified fatty acid levels between the groups (Figure 1e). Furthermore, serum estradiol levels did not significantly differ between the two groups (Figure 1f).

### 3.2. Isoflavone Mice Showed Higher Grip Strength and Lower Levels of Gene Expression Associated with Muscle Atrophy

Subsequently, assessment of the plantaris muscle was conducted. Representative images depicting HE-stained sections of the plantaris muscle are presented in Figure 2a. There were no significant differences in the cross-sectional areas of the plantaris muscles between the two groups (Figure 2b). The absolute and relative grip strengths in the isoflavone-treated mice were significantly higher than those in the control mice (Figure 2c). However, no significant differences were found in the absolute and relative weights of the plantaris and soleus muscles between the two groups (Figure 2d). Figure 2e displays heatmaps of the top 50 validated genes. These heatmaps were created using GSEA software (version 4.1.0), with the differential expression of upregulated (red) and downregulated (blue) genes within each chip scaled according to the specified color code. In the GSEA analyses, two hallmark gene sets were significantly enriched in control mice (Figure 2e). The most significantly enriched hallmark gene set was TNF-α signaling via NF-κB and unfolded protein response. We also analyzed the top 50 genes that were upregulated or downregulated in isoflavone-treated mice. In isoflavone mice, the expression of Trim63 was downregulated compared with that in control mice (Figure 2e). We examined protein expression levels in the gastrocnemius muscle. Using western blotting, we found that muscle-specific RING finger protein 1 (MuRF-1) signals were inactivated in the isoflavone mice (Figure 2f).

### 3.3. Administration of Isoflavone Changed the Gut Microbiota in OVX Mice

In the analysis of the gut microbiota, our initial focus was on assessing the relative abundances of different phyla across the two groups (Figure 3a). Compared with control mice, in isoflavone-treated mice, members of the phylum Bacteroidetes were more abundant while those of Firmicutes were less abundant (Figure 3c). OTUs, the Shannon index, Chao1, the Simpson index, and diversity indices exhibited markedly higher values in isoflavone-treated mice compared to control mice (Figure 3b). Seven taxa (e.g., genera Butyricimonas and Peptococcus) were overrepresented, and four taxa (e.g., the genus Pseudoflavonifractor) were underrepresented in the isoflavone mice (Figure 3d). A co-abundance network of OTUs was established in 12 samples derived from mice, concomitant with serum metabolomic data, aimed at further elucidating the relationship between CAGs and metabolic modules. Twelve distinct CAGs were identified. Notably, CAG5 and CAG7 exhibited markedly disparate levels of enrichment between the cohorts subjected to isoflavone treatment and those under control conditions (see Figure 3f). CAG5, which included the OTUs annotated as HQ681741, AB606312, HE607129, JF260132, JF254417, and AB443949 (accession numbers), was significantly enriched in isoflavones. Conversely, CAG7, including JF243907 (accession number), was significantly enriched in the control mice (Figure 3e,f).

### 3.4. Correlation of Serum Isoflavones with Gut Microbiota and Muscle Gene Cluster

We measured the concentrations of isoflavones, particularly genistein, daidzein, and their metabolite equol, in mouse serum. Daidzein, equol, and genistein showed differences in enrichment between the isoflavone and control mice (Figure 4a). The chemical structure of each is shown in Figure 4b. Daidzein and equol were significantly enriched in isoflavone mice. We then analyzed the correlations between 12 differential CAGs and three isoflavone metabolites, and expression of gene sets related to TNF-α signaling via NF-κB and unfolded protein response (Figure 4c). CAG5 positively correlated with daidzein and equol, whereas CAG7 negatively correlated with daidzein, equol, and genistein. In addition, gene sets related to TNF-α signaling via NF-κB and unfolded protein response were negatively associated with daidzein, equol, and genistein.

## 4. Discussion

Menopausal disorders are a cause of osteoporosis, metabolic decline, and worsening sarcopenia. Hormone replacement therapy is one treatment option for these changes, but it has also been shown that hormone therapy increases the risk of stroke and venous thrombosis in postmenopausal women [33]. For this reason, as an alternative to hormone replacement therapy, there is a lot of interest in the estrogenic effects of isoflavones, which can be obtained from food.

Soy is the most abundant natural source of isoflavones. During soy fermentation, the glucoside moiety is removed by the gut microbiota, resulting in the production of isoflavone aglycones [34]. Aglycones include genistein, daidzein, and glycitein, and gut microbiota convert daidzein into equol. These isoflavones act on estrogen receptors and have various effects. The gut microbiota influences the host organism, manifesting its impact through nutritional, metabolic, physiological, immune, and endocrine modalities. Disruptions in this microbial milieu are intricately linked to conditions such as obesity and metabolic syndromes. With the above background, we conducted research on the changes in the gut microbiota, the expression of muscle atrophy-related genes, and isoflavone-related metabolites in order to investigate the beneficial effects of isoflavones on the prevention of sarcopenia caused by menopausal women.

We observed that the ratio of Bacteroidetes increased while that of Firmicutes decreased in the gut microbiota of the isoflavone-treated mice. Previous reports on these two bacterial groups have shown that the relative proportion of the Bacteroidetes phylum decreases in obese individuals compared with that in lean individuals, and this ratio increases when body weight is reduced through low-calorie diets [35]. In our study, a significant difference in the Bacteroidetes/Firmicutes ratio of the gut microbiota was observed between isoflavone-treated and control mice, suggesting that isoflavone intake may alter the microbial profile in the gut microbiota.

Furthermore, in mice administered isoflavones, the variety of gut microbiota increased significantly, with CAG5 (HQ681741, AB606312, HE607129, JF260132, JF254417, AB443949 (accession number)) in particular increasing. There are few reports on the functions of these gut microbiota. However, CAG5, which was significantly increased in isoflavone mice, was positively correlated with increased blood equol–daidzein concentration. The present study suggests that the administration of isoflavones to OVX mice increases the gut microbiota, such as Bacteroides and CAG5, and promotes isoflavone metabolism and absorption, resulting in an increase in serum equol and daidzein concentrations.

GSEA analysis of soleus muscle cells indicated a reduction in the expression of TNF-α signaling mediated by NF-κB, as well as the unfolded protein response, in mice treated with isoflavones. One of the mechanisms underlying the onset of sarcopenia involves the activation of NF-κB by TNF-α signaling. TNF-α signaling suppresses the mRNA encoding of the myogenic transcription factor MyoD, thereby inhibiting the formation of new muscle fibers [36]. It is also considered that TNF-α signaling directly induces catabolism in the skeletal muscle, leading to muscle atrophy through the ubiquitin–proteasome system, which may be involved in the unfolded protein response [37]. These reports and the results of the present study suggest that isoflavone intake may suppress TNF-α signaling via NF-κB and the expression of the unfolded protein response, which may act to inhibit muscle atrophy. The present results also show a predominant decrease in muscle RING-finger protein-1 (MuRF-1) in isoflavone mice. MuRF-1 is a ubiquitin ligase (E3) involved in the degradation of muscle proteins and directs the polyubiquitination of proteins to target them for degradation by the 26S proteasome [38]. In previous reports, it has been reported that genistein and daidzein both exhibit inhibitory effects on TNF-α-induced MuRF-1 expression, and suppress muscle atrophy in myotube cells [17]. These results indicate that isoflavone intake inhibits proteolysis in the muscle tissue by suppressing MuRF-1 expression.

Previous reports have suggested that isoflavones have a positive effect on glucose tolerance and hypertension [12,13,14]. However, in the current study, despite observing an increase in serum equol and daidzein concentrations in isoflavone mice, no significant differences were found in serum markers related to glucose tolerance, lipid metabolism, or hepatic metabolism between isoflavone and control mice (Appendix A). Ovariectomy induces dramatic changes in sex hormone levels, resulting in significant metabolic changes. Therefore, we consider the possibility that external isoflavone intake is unlikely to be effective enough to significantly improve the metabolic abnormalities caused by hormonal changes. Therefore, isoflavone intake did not appear to significantly improve metabolic abnormalities induced by ovariectomy-related hormonal changes.

However, the results of this study showed that isoflavones cause changes in the intestinal bacterial flora and suppress the expression of muscle atrophy genes, with the latter leading to improvements in muscle quality and grip strength. In this study, 13-week-old mice were given 0.053 mg/g body weight of isoflavone. Based on BSA (body surface area) [39], this is equivalent to a daily dose of 14.42 mg for a 50 kg human. This is the equivalent of 20 g of natto or 100g of tofu. The amount of isoflavone consumed by the Japanese is relatively high, at 22.6~54.3 mg per day. Continuous intake of isoflavones may be effective in preventing muscle weakness in menopausal women.

In this study, in mice administered isoflavones, blood estradiol tended to increase, but this was not significant. A previous meta-analysis reported that soy and isoflavone intake caused a slight increase in estradiol in postmenopausal women, but this was also not significant [40], suggesting that isoflavones do not directly increase serum estradiol.

This research is a pioneering study that has demonstrated the effects of isoflavone intake on changes in the intestinal flora and the effects of isoflavone metabolites on muscle atrophy-related genes. However, there are some limitations. It is known that there are individual differences in gut microbiota, which play an important role in metabolism and the effects of isoflavones. However, detailed analysis of the gut microbiota of individual mice has not been investigated. Previous data on changes in the gut microbiota of the isoflavone-administered group (e.g., CAG5 and CAG7) are limited. Additional studies are required on gut microbiota. Moreover, it is essential to investigate whether high-dose isoflavone administration positively impacts glucose tolerance, lipid metabolism, and liver metabolism.

## 5. Conclusions

In summary, the intake of isoflavones may enhance the abundance of the gut microbiota, thereby promoting the metabolism and absorption of isoflavones, which subsequently alters gene expression. Therefore, there are potential implications for the suppression of muscle atrophy. These results expand our knowledge of the treatment of sarcopenia in menopausal women with impaired glucose tolerance, and pave the way for new strategies for the treatment and prevention of sarcopenia.

## Figures and Tables

**Figure 1 nutrients-16-03295-f001:**
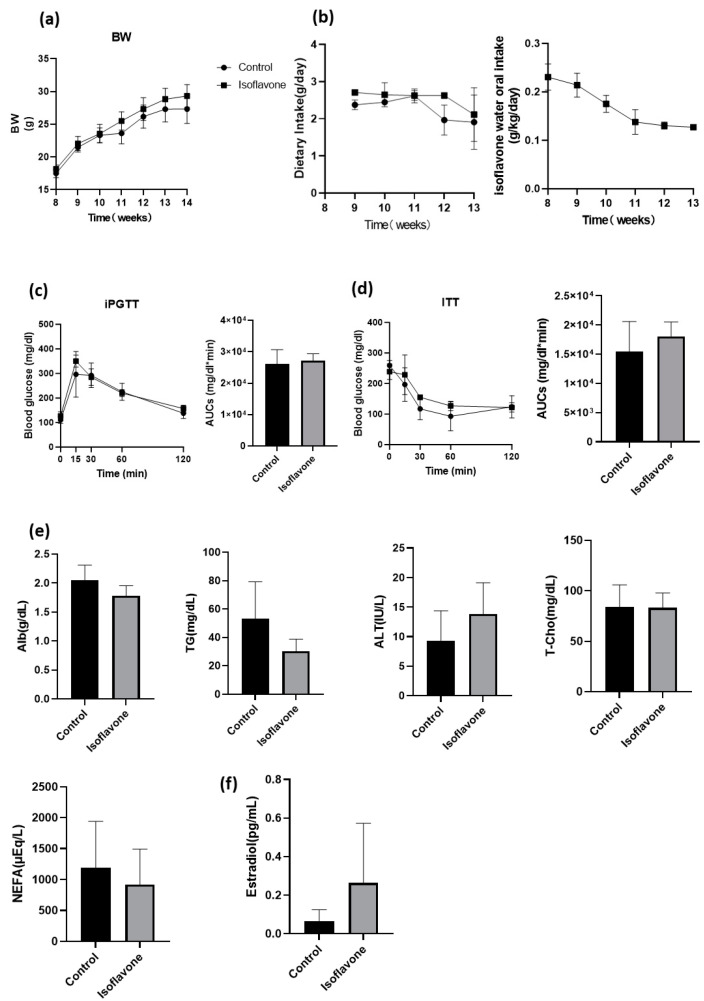
Administration of isoflavone did not change body weight, glucose tolerance, and lipid metabolism in OVX mice. (**a**) Body weight (*n* = 6); (**b**) change in dietary and isoflavone water intake (*n* = 6); (**c**) results from the intraperitoneal glucose tolerance test (iPGTT; 1 g/kg body weight) along with area under the curve (AUC) analysis (*n* = 6); (**d**) findings from the insulin tolerance test (ITT; 0.5 U/kg body weight), including AUC analysis (*n* = 6); (**e**) serum levels of albumin (Alb), alanine aminotransferase (ALT), triglycerides (TG), total cholesterol (T-Chol), and nonesterified fatty acids (NEFA) (*n* = 6); (**f**) serum estradiol levels (*n* = 6). Data are presented as mean ± SD values and were analyzed using a paired *t*-test. OVX: ovariectomy.

**Figure 2 nutrients-16-03295-f002:**
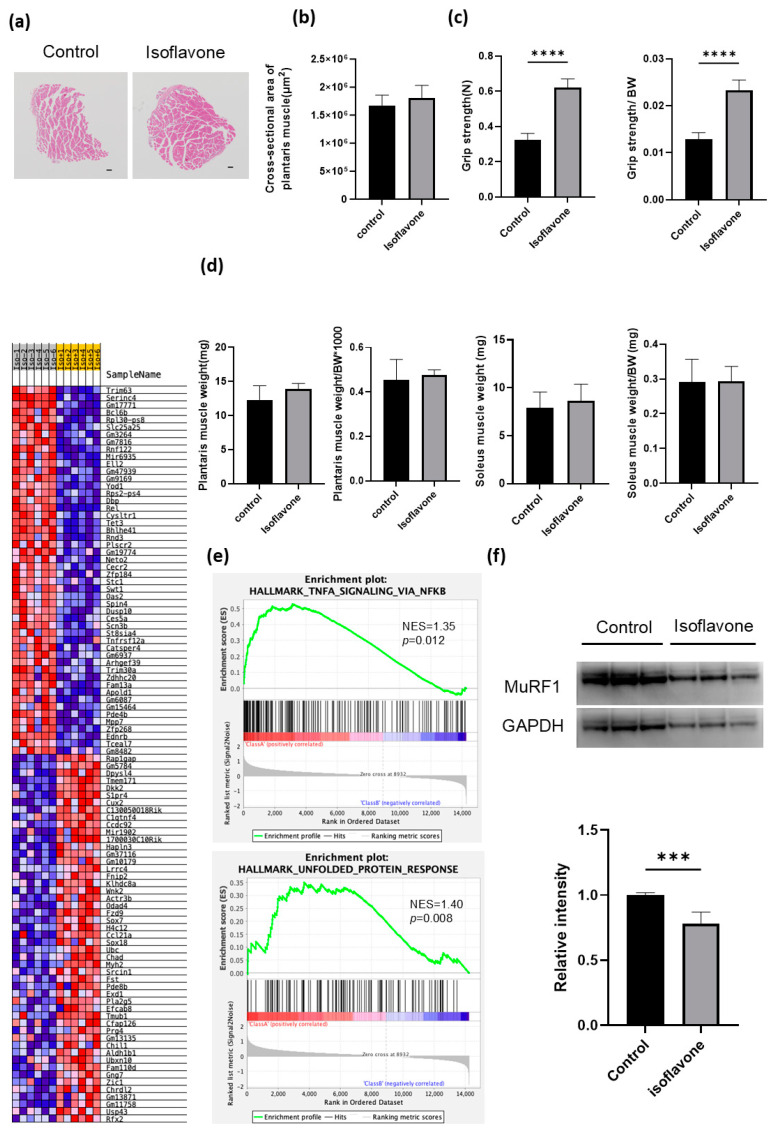
Isoflavone administration enhanced grip strength in mice and decreased the expression levels of muscle atrophy-related genes in OVX mice. (**a**) Shown are representative images of hematoxylin and eosin-stained sections of the plantaris muscle, collected from 14-week-old mice, with a scale bar of 100 μm; (**b**) the cross-sectional area and diameter of the plantaris muscle were measured (*n* = 6). (**c**) Both absolute and relative grip strength were recorded (*n* = 6); (**d**) the absolute and relative weights of the plantaris and soleus muscles were measured in 14-week-old mice (*n* = 6 in each group); (**e**) gene set enrichment analysis (GSEA) showed enrichment plots of three glycolysis-related gene sets, comparing isoflavone-treated mice with controls. A heatmap of 100 core genes is also provided for these comparisons; (**f**) western blot analysis detected MuRF1 levels in the gastrocnemius muscle, and the relative intensity of MuRF-1 was quantified (*n* = 6). Data are presented as mean ± SD values. Data were analyzed using a paired *t*-test. *** *p* < 0.001 and **** *p* < 0.0001.

**Figure 3 nutrients-16-03295-f003:**
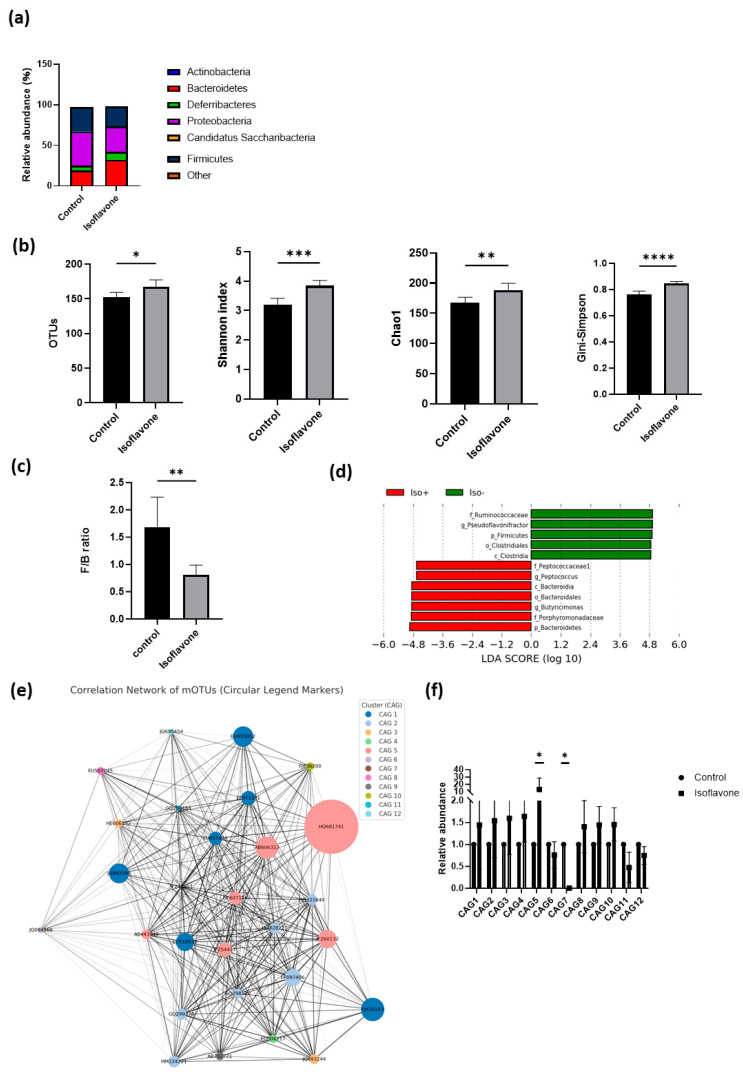
Administration of isoflavone changed the gut microbiota in OVX mice. (**a**) Constituents of the gut microbiota. Relative abundance of gut microbiota at the phylum levels (*n* = 3); (**b**) operational taxonomic units (OTUs) (*n* = 3), Shannon index, Chao1 (*n* = 3), and Gini–Simpson index (*n* = 3); (**c**) Firmicutes/Bacteroidetes ratio; (**d**) linear discriminant analysis (LDA) scores of gut microbiota of the control mice (green) and the isoflavone mice (red); (**e**) the structure of co-occurrence OTU networks is delineated. Nodes symbolize OTUs, while edges depict statistically significant positive correlations between each pair of OTUs. The dimensions of nodes reflect the relative abundance of OTUs within the dataset; (**f**) relative abundance of co-abundance gene groups (CAGs). Data are presented as mean ± SD values. * *p* < 0.05, ** *p* < 0.01, *** *p* < 0.001 and **** *p* < 0.0001.

**Figure 4 nutrients-16-03295-f004:**
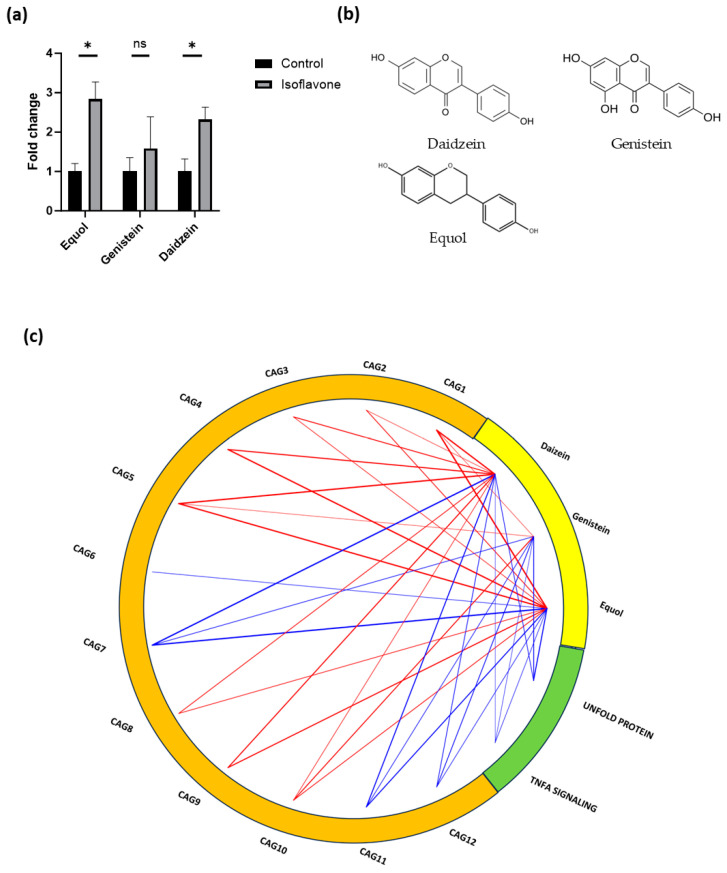
Correlation of serum isoflavones with intestinal microbiota and small intestinal gene cluster. (**a**) Fold change of concentration in serum of equol, genistein, and daidzein; (**b**) chemical structure of daidzein, genistein and equol; (**c**) circos plot showing the gene expression in relation to 12 differential CAGs and three isoflavone metabolites, and gene set related TNF-α signaling via NF-κB and unfolded protein response. Only pairs with an absolute value of correlation coefficient greater than 0.3 are connected at the edges. Red links stand for positive correlations and blue links stand for negative correlations. Data are presented as mean ± SD values. * *p* < 0.05. ns: not significant.

## Data Availability

The datasets were uploaded at https://www.ncbi.nlm.nih.gov/bioproject/PRJNA1154784 (accessed on 30 August 2024).

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
