# Peer review of "Effect of Isoflavone on Muscle Atrophy in Ovariectomized Mice"

_nutrients, 2024, doi:10.3390/nu16193295_

Round 1

Reviewer 1 Report

Comments and Suggestions for Authors

The report by Kawai et al. on isoflavone influence on muscle atrophy in OVX mice has errors that must be addressed.

1.Title is misleading,  This was an animal study

Please change to: Effect of isoflavone on muscle atrophy in ovariectomized mice

2. Introduction, 1st sentence, move to line 37

3. Line 40, reference 3 is a poor citation.  Note: estrogen levels decline before menopause, see reference: Lephart & Naftolin, 2021,11: 53-69, Dermatology and Therapy.

4. Section 2.1 there is no description of the isoflavone(s) used in this study?

What isoflavone(s) were used, what was the original concentration of this material? How was this isoflavone compound made to be water soluble?

5.  The Research Diets Inc D12327 has 45 grams of soybean oil (as fat), what is the concentration of isoflavones in this portion of the diet and how was this accounted for in the study?

6. Starting on line 143 thru 178 it describes how the serum isoflavones were analyzed.  Why was this data not reported?

Please answer the following questions:

What type of isoflavones were detected?

Please generate a figure with the chemical structures of the isoflavone compounds in the serum.

7. Section 2.12  the statistics are unclear, please describe what statistical test was used and were different statistical analyses used for different data?

8. in Figure 1F  why did the estradiol levels increase in the isoflavone treatment?

9. The discussion is lacking because the isoflavone treatment is never described, very vague.

10.  The summary is lacking/weak due to NOT understanding what the isoflavone treatment was used in this study.

11.  The true significance of this study is unknown based upon the poor experimental design/description of the methods and lack of reporting the results properly.

Author Response

Dear Editors and Reviewers

Thank you very much for reviewing our manuscript and offering valuable advice.

We have addressed your comments with point-by-point responses, and revised the manuscript accordingly.

1.Title is misleading,  This was an animal study

Please change to: Effect of isoflavone on muscle atrophy in ovariectomized mice

[Response]

Thank you for your excellent suggestion. As you indicated, we have revised the title as follows.

“Effect of isoflavone on muscle atrophy in ovariectomized mice”

  1. Introduction, 1stsentence, move to line 37

[Response]

Thank you very much for your excellent suggestion. In line with your suggestions, we have made the necessary revisions to the manuscript.

(1. Introduction Line34-39)

“Sarcopenia refers to a condition in which muscle mass decreases due to aging or other factors. Its prevalence is estimated to be 10% of the world’s population in both men and women worldwide [1]. Diabetes is a well-established risk factor for sarcopenia. Diabetes is also known to accelerate age-related decline in skeletal muscle mass and function [2]. The incidence of type 2 diabetes is increasing, and as the population ages, advances in its treatment and prevention of complications is imperative.”

  1. Line 40, reference 3 is a poor citation.  Note: estrogen levels decline before menopause, see reference: Lephart & Naftolin, 2021,11: 53-69, Dermatology and Therapy.

[Response]

Thank you for the wonderful suggestion. We have reviewed and corrected the reference to the relationship between menopause, sarcopenia, and estrogen.

  • Introduction Line 40-42)

“Menopause is caused by age-related changes in estrogen levels [3], and decreased estrogen levels are recognized as an exacerbating factor for sarcopenia [4].”

Reference [3]:

Geraci, A.; Calvani, R.; Ferri, E.; Marzetti, E.; Arosio, B.; Cesari, M. Sarcopenia and Menopause: The Role of Estradiol. Front Endocrinol (Lausanne) 2021, 12, doi:10.3389/FENDO.2021.682012.

Reference [4]:

Lu, L.; Tian, L. Postmenopausal Osteoporosis Coexisting with Sarcopenia: The Role and Mechanisms of Estrogen. J En-docrinol 2023, 259, doi:10.1530/JOE-23-0116.

  1. Section 2.1 there is no description of the isoflavone(s) used in this study?

What isoflavone(s) were used, what was the original concentration of this material? How was this isoflavone compound made to be water soluble?

[Response]

Thank you for your suggestion. We have included a more detailed description of the soy isoflavones we used. We used Soyaflavone HG provided by the Fuji Foundation for Protein Research. We have added its composition.

(2. Materials and Methods Line 86-95)

“The soy isoflavones have an isoflavone content of 40.74% (Soyaflavone HG; Fuji Foundation for Protein Research), and the composition of isoflavones in Soyaflavone HG consists of various forms of isoflavones measured in grams per kilogram (g/kg). The most abundant isoflavones are malonyl daidzin, which is present at 361.7 g/kg, and daidzin at 239.4 g/kg. Other significant isoflavones include glycitin at 95.9 g/kg, malonyl glycitin at 135.9 g/kg, and genistin at 73.4 g/kg. Less abundant forms, such as acetyl daidzin (10.7 g/kg), acetyl glycitin (9.3 g/kg), and acetyl genistin (2.5 g/kg), are also present. Additionally, trace amounts of the aglycones daidzein (1.8 g/kg), genistein (0.1 g/kg), and glycitein (0.5 g/kg) are found in the composition. Soyaflavone HG was dissolved in water to a concentration of 0.1%.”

  1. The Research Diets Inc D12327 has 45 grams of soybean oil (as fat), what is the concentration of isoflavones in this portion of the diet and how was this accounted for in the study?

[Response]

Thank you for your suggestion. As you pointed out, The Research Diets Inc D12327 contains soybean oil. According to previous reports, the amount of isoflavones contained in soybean oil is very small, so we do not think that the amount of isoflavones ingested from HFHSD intake needs to be a concern in this case.

(2. Materials and Methods  Line 100-103)

“The HFHSD used in this study contained soybean oil. However, previous reports have shown that the amount of isoflavones contained in soybean oil is very small, so there is no need to be concerned about the amount of isoflavones ingested from HFHSD [24]. “

Reference [24]:

Rizzo, G.; Baroni, L. Soy, Soy Foods and Their Role in Vegetarian Diets. Nutrients 2018, 10, doi:10.3390/NU10010043.

  1. Starting on line 143 thru 178 it describes how the serum isoflavones were analyzed.  Why was this data not reported?

Please answer the following questions:

What type of isoflavones were detected? Please generate a figure with the chemical structures of the isoflavone compounds in the serum.

[Response]

Thank you for your excellent suggestion. The results of isoflavone concentration in mouse serum are shown in Figure 4a. We apologize for the confusion. Three isoflavones were detected: daidzein, equol, and genistein. The chemical structure of each has been added in Figure 4b.

(3. Results Line 364-367)

“We measured the concentrations of isoflavones, particularly genistein, daidzein, and their metabolite equol, in mouse serum. Daidzein, equol, and genistein showed differences in enrichment between the isoflavone and control mice (Figure 4a). The chemical structure of each is shown Figure 4b.”

  1. Section 2.12  the statistics are unclear, please describe what statistical test was used and were different statistical analyses used for different data?

[Response]

Thank you for your suggestion. We have added a more detailed analysis method.

(2. Materials and Methods Line 266-270)

“Data underwent analysis using GraphPad Prism Version 10 (GraphPad Software, La Jolla, CA, USA). Welch’s t-test was applied for comparisons between two groups. The findings were delineated as the means ± standard deviations (SD). P values below 0.05 were considered statistically significant. Figures were generated using GraphPad Prism Version 10.”

  1. in Figure 1F  why did the estradiol levels increase in the isoflavone treatment?

[Response]

Thank you for your suggestion. We observed an increasing trend in serum estradiol concentrations in the isoflavone-treated group, but this did not differ significantly.

To the best of our knowledge, different studies have reported different results on whether isoflavone intake affects blood estradiol concentrations, and the results are not consistent.

We have added a note on this point.

 (4. Discussion Line 455-458)

“In this study, in mice administered isoflavones, blood estradiol tended to increase, but this was not significant. A previous meta-analysis reported that soy and isoflavone intake caused a slight increase in estradiol in postmenopausal women, but this was also not significant [40], suggesting that isoflavones do not directly increase serum estradiol.”

Reference [40]:

Hooper, L.; Ryder, J.J.; Kurzer, M.S.; Lampe, J.W.; Messina, M.J.; Phipps, W.R.; Cassidy, A. Effects of Soy Protein and Isoflavones on Circulating Hormone Concentrations in Pre- and Post-Menopausal Women: A Systematic Review and Meta-Analysis. Hum Reprod Update 2009, 15, 423–440, doi:10.1093/HUMUPD/DMP010.

  1. The discussion is lacking because the isoflavone treatment is never described, very vague.

[Response]

Thank you for your suggestion. As you suggested, I have added a discussion of isoflavone therapy with references. 

(4. Discussion  Line 385-390)

“Menopausal disorders are a cause of osteoporosis, metabolic decline, and worsening sarcopenia. Hormone replacement therapy is one treatment option for these changes, but it has also been shown that hormone therapy increases the risk of stroke and venous thrombosis in postmenopausal women [33]. For this reason, as an alternative to hormone replacement therapy, there is a lot of interest in the estrogenic effects of isoflavones, which can be obtained from food.”

Reference [33]:

Gu, Y.; Han, F.; Xue, M.; Wang, M.; Huang, Y. The Benefits and Risks of Menopause Hormone Therapy for the Cardiovascular System in Postmenopausal Women: A Systematic Review and Meta-Analysis. BMC Womens Health 2024, 24, doi:10.1186/S12905-023-02788-0.

  1. The summary is lacking/weak due to NOT understanding what the isoflavone treatment was used in this study.

[Response]

Thank you for your suggestion. As you have indicated, we will add a more detailed explanation of the administration of isoflavones. (2. Materials and Methods Line 86-95)

Also, The isoflavone dosage in this study was converted to a human dosage and discussed. 

(4. Discussion Line 447-454)

“However, the results of this study showed that isoflavones cause changes in the intestinal bacterial flora, suppress the expression of muscle atrophy genes, and this leads to improvements in muscle quality and grip strength. In this study, 13-week-old mice were given 0.053 mg/g body weight of isoflavone. Based on BSA (body surface area) [39], this is equivalent to a daily dose of 14.42mg for a 50kg human. This is the equivalent of 20g of natto or 100g of tofu. The amount of isoflavone consumed by the Japanese is relatively high, at 22.6~54.3mg per day. Continuous intake of isoflavones may be effective in preventing muscle weakness in menopausal women.”

Reference [39]:

Reagan‐Shaw, S.; Nihal, M.; Ahmad, N. Dose Translation from Animal to Human Studies Revisited. FASEB J 2008, 22, 659–661, doi:10.1096/FJ.07-9574LSF.

In addition, we have included a summary of this research in the conclusion. 

(5. Conclusion Line 471-476)

“In summary, the intake of isoflavones may enhance the abundance of gut microbiota, thereby promoting the metabolism and absorption of isoflavones, which subsequently alters gene expression. Therefore, there are potential implications for the suppression of muscle atrophy. These results expand our knowledge of the treatment of sarcopenia in menopausal women with impaired glucose tolerance, and pave the way for new strategies for the treatment and prevention of sarcopenia.”

  1. The true significance of this study is unknown based upon the poor experimental design/description of the methods and lack of reporting the results properly.

[Response]

Thank you for your excellent suggestion. We have added more information on the experimental plan and methods, including information on rearing methods, sample sizes and feeding methods. We have also added information on the materials 

(2. Methods  Line 72-115)

“All animal experiments were conducted following approval from the Animal Research Committee at Kyoto Prefectural University of Medicine (M2023-78). Female C57BL6/J mice, 6 weeks old, weighing between 18 and 20 grams (n = 12), were sourced from SHIMIZU Laboratory Supplies Co., Ltd. (Kyoto, Japan). The mice were maintained in the animal care facility of Kyoto Prefectural University of Medicine under specific pathogen-free conditions, with the temperature controlled at 23 ± 1.5°C, and exposed to a 12-hour light/dark cycle (7 a.m. to 7 p.m.). Mice were housed in cages of W220 × L320×H135(mm), with six mice in each cage. During the first week, the mice were acclimatized, and at 7 weeks of age, all mice were ovariectomized by administering an anesthetic combination of 0.3 mg/kg medetomidine, 4.0 mg/kg midazolam, and 5.0 mg/kg butorphanol to produce an estrogen-deficient state. At 8 weeks of age, OVX mice were randomly assigned to two groups: one receiving a high-fat, high-sucrose diet (HFHSD, 459 kcal/100 g, 20 % protein, 40 % carbohydrate, and 40 % fat; D12327, Research Diets, Inc., New Brunswick, NJ, USA) and soy isoflavone water (0.1%), the other receiving HFHSD and normal water for 6 weeks (n=6 per group). The soy isoflavones have an isoflavone content of 40.74% (Soyaflavone HG; Fuji Foundation for Protein Research), and the composition of isoflavones in Soyaflavone HG consists of various forms of isoflavones measured in grams per kilogram (g/kg). The most abundant isoflavones are malonyl daidzin, which is present at 361.7 g/kg, and daidzin at 239.4 g/kg. Other significant isoflavones include glycitin at 95.9 g/kg, malonyl glycitin at 135.9 g/kg, and genistin at 73.4 g/kg. Less abundant forms, such as acetyl daidzin (10.7 g/kg), acetyl glycitin (9.3 g/kg), and acetyl genistin (2.5 g/kg), are also present. Additionally, trace amounts of the aglycones daidzein (1.8 g/kg), genistein (0.1 g/kg), and glycitein (0.5 g/kg) are found in the composition. Soyaflavone HG was dissolved in water to a concentration of 0.1%. Previous reports have shown that mice fed the HFHSD showed signs of sarcopenia [22], and is closely related to worsening insulin resistance and inflammation of muscle tissue [23]. These mice are a model for the elderly with metabolic diseases due to the effects of modern Westernized diets, and we thought they would be suitable for studying the effects of isoflavones on menopausal women. For this reason, we chose the HFHSD. The HFHSD used in this study contained soybean oil. However, previous reports have shown that the amount of isoflavones contained in soybean oil is very small, so there is no need to be concerned about the amount of isoflavones ingested from HFHSD [24]. In order to ensure that pair feeding was carried out, the same amount of food was given to each group, and the weight and amount of food and water consumed were measured twice a week. Power analysis was conducted based on the mean and standard deviation of the area under the curve (AUC) of the intraperitoneal glucose tolerance test (IPGTT) performed at 13 weeks of age in the two groups. With a significance level of 0.05 and a power of 80%, it was calculated that a minimum of six biological replicates per group would be required to detect a statistically significant difference. Consequently, the experiment was carried out using six animals per group (n=6). When the mice reached 14 weeks of age, after fasting for 16 hours, they were euthanized by administration of a combination anesthetic of 0.3 mg/kg medetomidine, 4.0 mg/kg midazolam, and 5.0 mg/kg butorphanol. All investigators were not blinded to the experimental conditions. There were no mice in any of the experimental groups that were not included in the analysis.”

Reference [22]:

Burchfield, J.G.; Kebede, M.A.; Meoli, C.C.; Stöckli, J.; Whitworth, P.T.; Wright, A.L.; Hoffman, N.J.; Minard, A.Y.; Ma, X.; Krycer, J.R.; et al. High Dietary Fat and Sucrose Results in an Extensive and Time-Dependent Deterioration in Health of Multiple Physiological Systems in Mice. J Biol Chem 2018, 293, 5731–5745, doi:10.1074/JBC.RA117.000808.

Reference [23]:

Rasool, S.; Geetha, T.; Broderick, T.L.; Babu, J.R. High Fat With High Sucrose Diet Leads to Obesity and Induces Myo-degeneration. Front Physiol 2018, 9, doi:10.3389/FPHYS.2018.01054.

Reference [24]:

Rizzo, G.; Baroni, L. Soy, Soy Foods and Their Role in Vegetarian Diets. Nutrients 2018, 10, doi:10.3390/NU10010043.

The results of this study have been summarized, and the need for further research on the application to menopausal women and future prospects has been added.

(4. Discussion Line 459-468)

This research is a pioneering study that has demonstrated the effects of isoflavone intake on changes in the intestinal flora and the effects of isoflavone metabolites on anti-muscle atrophy genes. However, there are some limitations. It is known that there are individual differences in gut microbiota, which plays an important role in metabolism and the effects of isoflavones. However, detailed analysis of the gut microbiota of individual mice has not been investigated. There is little previous data on the gut microbiota contained in the isoflavone-administered group that changed (e.g. CAG5 and CAG7). Additional studies are required on the gut microbiota. Moreover, it is essential to investigate whether high-dose isoflavone administration positively impacts glucose tolerance, lipid metabolism, and liver metabolism.

Reviewer 2 Report

Comments and Suggestions for Authors

The authors report on the effects of isoflavone intake on muscle integrity in OVX mice. Secondary outcomes are the effects of isoflavones on HFHS-induced disturbance of glucose and lipid metabolism. The purpose of evaluating these distinct outcomes in one study needs further explanation. Some methods also need greater description.

Title: Needs to specify mice. Suggest " Investigation of the effect of isoflavone on muscle atrophy in ovariectomized mice."

Introduction: Line 60-61 cites one article whereas you refer to "numerous reports." Please list other studies. 

Methods: Line 78 - specify which group was pair-fed to which group and the procedure of pair-feeding. Describe food intake measurements. Was water intake measured?

Explain why a HFHS diet was fed. Is there evidence that this diet induces sarcopenia? Would the OVX mice develop sarcopenia on a normal diet? It seems that trying to test the effect of isoflavones on muscle atrophy was obscured by also testing the effect of isoflavones on HFHS-induced disorders of glucose and lipid metabolism. This needs to be clearly discussed in the paper.

Line 88, 92: Describe how blood was collected for analysis.

Line 108: Grip-strength testing must be described in detail.

Line 180: When were feces collected (week 14?) You were able to collect feces from the appendix in 16-hr-fasted mice?

Results: Line 238: Clarify that mice were pair-fed, so a difference in body weight at the end of the experiment would not be expected.

Figure 1b: Again, the procedure of food intake measurement needs to be explained in Methods.

Line 256: There is no need to define the * ** *** if no differences were seen.

Line 277: This description of MuRF-1 should be provided in Methods to explain why it was evaluated.

The reference list is limited in current research - most references are >10 years old.  Newer studies should be cited such as: https://doi.org/10.3389/fnut.2024.1429242

Author Response

Dear Editors and Reviewers

Thank you very much for reviewing our manuscript and offering valuable advice.
We have addressed your comments with point-by-point responses, and revised the manuscript accordingly.

The authors report on the effects of isoflavone intake on muscle integrity in OVX mice. Secondary outcomes are the effects of isoflavones on HFHS-induced disturbance of glucose and lipid metabolism. The purpose of evaluating these distinct outcomes in one study needs further explanation. Some methods also need greater description.

[Response]

Thank you for your excellent suggestion.

I have added a note about the purpose of evaluating the secondary results of this study, as well as the reason why I used HFHSD.

(2. Materials and Methods  Line 95-100)

“Previous reports have shown that mice fed the HFHSD showed signs of sarcopenia [22], and is closely related to worsening insulin resistance and inflammation of muscle tissue [23]. These mice are a model for the elderly with metabolic diseases due to the effects of modern Westernized diets, and we thought they would be suitable for studying the effects of isoflavones on menopausal women. For this reason, we chose the HFHSD.”

Reference [22]:

Burchfield, J.G.; Kebede, M.A.; Meoli, C.C.; Stöckli, J.; Whitworth, P.T.; Wright, A.L.; Hoffman, N.J.; Minard, A.Y.; Ma, X.; Krycer, J.R.; et al. High Dietary Fat and Sucrose Results in an Extensive and Time-Dependent Deterioration in Health of Multiple Physiological Systems in Mice. J Biol Chem 2018, 293, 5731–5745, doi:10.1074/JBC.RA117.000808.

Reference [23]:

Rasool, S.; Geetha, T.; Broderick, T.L.; Babu, J.R. High Fat With High Sucrose Diet Leads to Obesity and Induces Myo-degeneration. Front Physiol 2018, 9, doi:10.3389/FPHYS.2018.01054.

Title: Needs to specify mice. Suggest " Investigation of the effect of isoflavone on muscle atrophy in ovariectomized mice."

[Response]

Thank you for your excellent suggestion. As you indicated, we have revised the title as follows.

“Effect of isoflavone on muscle atrophy in ovariectomized mice”

Introduction: Line 60-61 cites one article whereas you refer to "numerous reports." Please list other studies. 

[Response]

Thank you for pointing this out. I have added the references. Reference 19,20,21

Reference [19]:

Siddharth, J.; Chakrabarti, A.; Pannérec, A.; Karaz, S.; Morin-Rivron, D.; Masoodi, M.; Feige, J.N.; Parkinson, S.J. Aging and Sarcopenia Associate with Specific Interactions between Gut Microbes, Serum Biomarkers and Host Physiology in Rats. Aging 2017, 9, 1698–1720, doi:10.18632/AGING.101262.

Reference [20]:

Kang, L.; Li, P.; Wang, D.; Wang, T.; Hao, D.; Qu, X. Alterations in Intestinal Microbiota Diversity, Composition, and Function in Patients with Sarcopenia. Sci Rep 2021, 11, doi:10.1038/S41598-021-84031-0.

Reference [21]:

Song, Q.; Zhu, Y.; Liu, X.; Liu, H.; Zhao, X.; Xue, L.; Yang, S.; Wang, Y.; Liu, X. Changes in the Gut Microbiota of Patients with Sarcopenia Based on 16S RRNA Gene Sequencing: A Systematic Review and Meta-Analysis. Front Nutr 2024, 11, doi:10.3389/FNUT.2024.1429242.

Methods: Line 78 - specify which group was pair-fed to which group and the procedure of pair-feeding. Describe food intake measurements. Was water intake measured?

[Response]

Thank you for your comments. I have revised the pair-feeding procedure. We measured how intake isoflavone water in the isoflavone group. This result has been added to Figure 1b.

(2. Materials and Methods  Line 75-86).

“The mice were maintained in the animal care facility of Kyoto Prefectural University of Medicine under specific pathogen-free conditions, with the temperature controlled at 23 ± 1.5°C, and exposed to a 12-hour light/dark cycle (7 a.m. to 7 p.m.). Mice were housed in cages of W220 × L320×H135, with six mice in each cage. During the first week, the mice were acclimatized, and at 7 weeks of age, all mice were ovariectomized by administering an anesthetic combination of 0.3 mg/kg medetomidine, 4.0 mg/kg midazolam, and 5.0 mg/kg butorphanol to produce an estrogen-deficient state. At 8 weeks of age, OVX mice were randomly assigned to two groups: one receiving a high-fat, high-sucrose diet (HFHSD, 459 kcal/100 g, 20 % protein, 40 % carbohydrate, and 40 % fat; D12327, Research Diets, Inc., New Brunswick, NJ, USA) and soy isoflavone water (0.1%), the other receiving HFHSD and normal water for 6 weeks (n=6 per group).”

(2. Materials and Methods  Line 103-105).

“In order to ensure that pair feeding was carried out, the same amount of food was given to each group, and the weight and amount of food and water consumed were measured twice a week.”

Explain why a HFHS diet was fed. Is there evidence that this diet induces sarcopenia? Would the OVX mice develop sarcopenia on a normal diet? It seems that trying to test the effect of isoflavones on muscle atrophy was obscured by also testing the effect of isoflavones on HFHS-induced disorders of glucose and lipid metabolism. This needs to be clearly discussed in the paper.

[Response]

Thank you for your excellent suggestion. I have added a note about the reason why I used HFHSD.

(2. Materials and Methods  Line 95-100)

“Previous reports have shown that mice fed the HFHSD showed signs of sarcopenia [22], and is closely related to worsening insulin resistance and inflammation of muscle tissue [23]. These mice are a model for the elderly with metabolic diseases due to the effects of modern Westernized diets, and we thought they would be suitable for studying the effects of isoflavones on menopausal women. For this reason, we chose the HFHSD.”

Reference [22]:

Burchfield, J.G.; Kebede, M.A.; Meoli, C.C.; Stöckli, J.; Whitworth, P.T.; Wright, A.L.; Hoffman, N.J.; Minard, A.Y.; Ma, X.; Krycer, J.R.; et al. High Dietary Fat and Sucrose Results in an Extensive and Time-Dependent Deterioration in Health of Multiple Physiological Systems in Mice. J Biol Chem 2018, 293, 5731–5745, doi:10.1074/JBC.RA117.000808.

Reference [23]:

Rasool, S.; Geetha, T.; Broderick, T.L.; Babu, J.R. High Fat With High Sucrose Diet Leads to Obesity and Induces Myo-degeneration. Front Physiol 2018, 9, doi:10.3389/FPHYS.2018.01054.

Line 88, 92: Describe how blood was collected for analysis.

[Response]

Thank you for your excellent suggestion. The blood was collected from the cardiac cavity. This has been corrected.

(2. Materials and Methods  Line124)

“Immediately after euthanasia, blood was collected by cardiac puncture.”

Line 108: Grip-strength testing must be described in detail.

[Response]

Thank you for pointing this out. We have added more details about grip-strength measurement.

(2. Materials and Methods  Line 141-144)

“To measure grip strength, we used a mouse grip strength meter (model DS2-50N, Imada Manufacturing Co., Ltd., Toyohashi City, Aichi Prefecture) and measured grip strength at 14 weeks of age. We performed three consecutive measurements at one-minute intervals and standardized grip strength according to body weight. “

Line 180: When were feces collected (week 14?) You were able to collect feces from the appendix in 16-hr-fasted mice?

[Response]

Thank you for pointing this out. The appendiceal feces was collected immediately after the 14-week euthanasia.

(2. Materials and Methods  Line 217-218)

“Immediately after euthanasia, fecal samples were meticulously obtained from the appendix and subsequently transferred to cryotubes.”

 Our research group usually analyzes intestinal flora using appendiceal stool. As the mouse grew, feces accumulated in the appendix for long periods of time, so it is possible to collect appendiceal feces even after fasting for 16 hours.

Results: Line 238: Clarify that mice were pair-fed, so a difference in body weight at the end of the experiment would not be expected.

[Response]

Thank you for pointing that out. I've added a note about that.

(3. Results  Line 274-276)

“Since the mice were fed in pairs, no discernible differences were noted in the body weights of mice administered HFHSD with isoflavone (isoflavone mice) compared to those given HFHSD without isoflavone (control mice) (Figure 1a).”

Figure 1b: Again, the procedure of food intake measurement needs to be explained in Methods.

[Response]

Thank you for pointing this out. We have added more details about this. (2. Materials and Methods  Line 75-86, Line 103-105).

Line 256: There is no need to define the * ** *** if no differences were seen.

[Response]

Thank you for pointing this out. It has been deleted.

Line 277: This description of MuRF-1 should be provided in Methods to explain why it was evaluated.

[Response]

Thank you for pointing this out. We have added an explanation of MuRF-1 to the Methods section.

The reference list is limited in current research - most references are >10 years old. 

Newer studies should be cited such as: https://doi.org/10.3389/fnut.2024.1429242

[Response]

Thank you for pointing this out. We have added new research from within the last 10 years.

(reference 3, 16, 20,21,22,23,24)

Reference [3]:

Geraci, A.; Calvani, R.; Ferri, E.; Marzetti, E.; Arosio, B.; Cesari, M. Sarcopenia and Menopause: The Role of Estradiol. Front Endocrinol (Lausanne) 2021, 12, doi:10.3389/FENDO.2021.682012.

Reference [16]:

Kitamura, K.; Erlangga, J.S.; Tsukamoto, S.; Sakamoto, Y.; Mabashi-Asazuma, H.; Iida, K. Daidzein Promotes the Expression of Oxidative Phosphorylation- and Fatty Acid Oxidation-Related Genes via an Estrogen-Related Receptor α Pathway to Decrease Lipid Accumulation in Muscle Cells. J Nutr Biochem 2020, 77, doi:10.1016/J.JNUTBIO.2019.108315.

Reference [20]:

Kang, L.; Li, P.; Wang, D.; Wang, T.; Hao, D.; Qu, X. Alterations in Intestinal Microbiota Diversity, Composition, and Function in Patients with Sarcopenia. Sci Rep 2021, 11, doi:10.1038/S41598-021-84031-0.

Reference [21]:

Song, Q.; Zhu, Y.; Liu, X.; Liu, H.; Zhao, X.; Xue, L.; Yang, S.; Wang, Y.; Liu, X. Changes in the Gut Microbiota of Patients with Sarcopenia Based on 16S RRNA Gene Sequencing: A Systematic Review and Meta-Analysis. Front Nutr 2024, 11, doi:10.3389/FNUT.2024.1429242.

Reference [22]:

Burchfield, J.G.; Kebede, M.A.; Meoli, C.C.; Stöckli, J.; Whitworth, P.T.; Wright, A.L.; Hoffman, N.J.; Minard, A.Y.; Ma, X.; Krycer, J.R.; et al. High Dietary Fat and Sucrose Results in an Extensive and Time-Dependent Deterioration in Health of Multiple Physiological Systems in Mice. J Biol Chem 2018, 293, 5731–5745, doi:10.1074/JBC.RA117.000808.

Reference [23]:

Rasool, S.; Geetha, T.; Broderick, T.L.; Babu, J.R. High Fat With High Sucrose Diet Leads to Obesity and Induces Myodegeneration. Front Physiol 2018, 9, doi:10.3389/FPHYS.2018.01054.

Reference [24]:

Rizzo, G.; Baroni, L. Soy, Soy Foods and Their Role in Vegetarian Diets. Nutrients 2018, 10, doi:10.3390/NU10010043.

Round 2

Reviewer 1 Report

Comments and Suggestions for Authors

The authors address all the items in the evaluation/review

Author Response

Thank you for taking your valuable time to review our  manuscript. Your recommendations have improved the quality of our manuscript.

Reviewer 2 Report

Comments and Suggestions for Authors

The authors have addressed most of my concerns and the manuscript is improved. Remaining comments:

The authors misuse the term "pair-feeding." Oxford reference definition: Restricting the intake of a group of control animals to match that of those receiving an experimental diet, so as to eliminate differences due to total amount of food consumed.

Lines 102, 273: Please remove the term pair feeding.

Line 273: Due to the lack of pair-feeding, you cannot comment on the effect of food provision on body weight - the mice ate as much as they desired. It seems correct to state that they were fed ad libitum. The authors can comment only on the relationship between food intake and body weight.

State the results of comparing food intake between the groups.

A limitation to add is that pair-feeding was not implemented. This would have ensured that outcomes were not affected by differences in food intake.

Author Response

Dear Editors and Reviewers

Thank you very much for reviewing our manuscript and offering valuable advice.

We have responded to the comments we received and revised the manuscript.

The authors misuse the term "pair-feeding." Oxford reference definition: Restricting the intake of a group of control animals to match that of those receiving an experimental diet, so as to eliminate differences due to total amount of food consumed.

Lines 102, 273: Please remove the term pair feeding.

Line 273: Due to the lack of pair-feeding, you cannot comment on the effect of food provision on body weight - the mice ate as much as they desired. It seems correct to state that they were fed ad libitum. The authors can comment only on the relationship between food intake and body weight.

[Response]

Thank you for your excellent suggestion.

We apologize for not being able to explain the feeding method well. We carried out pair feeding in the following way.

We first obtained mice in the control group (6 weeks old, HFHSD + normal water) (n=6), and continued to measure their intake twice a week. After one week, we purchased mice in the isoflavone group (6 weeks old, HFHSD + 0.1% isoflavone water) (n=6). We gave the isoflavone group the same amount as the control group consumed at the same age.

We used this method for pair feeding. We believe that this is not free feeding in this study.

We apologize for not being able to explain the feeding method well.

We have revised it again.

(Line 72-89)

“All animal experiments were conducted following approval from the Animal Research Committee at Kyoto Prefectural University of Medicine (M2023-78). Female C57BL6/J mice, 6 weeks old, weighing between 18 and 20 grams (n = 12), were sourced from SHIMIZU Laboratory Supplies Co., Ltd. (Kyoto, Japan). We obtained mice in groups of six at weekly intervals. The mice were maintained in the animal care facility of Kyoto Prefectural University of Medicine under specific pathogen-free conditions, with the temperature controlled at 23 ± 1.5°C, and exposed to a 12-hour light/dark cycle (7 a.m. to 7 p.m.). Mice were housed in cages of W220 × L320×H135(mm), with six mice in each cage. During the first week, the mice were acclimatized, and at 7 weeks of age, all mice were ovariectomized by administering an anesthetic combination of 0.3 mg/kg medetomidine, 4.0 mg/kg midazolam, and 5.0 mg/kg butorphanol to produce an estrogen-deficient state.

From 8 weeks of age, the first six OVX mice were fed a high-fat, high-sucrose diet (HFHSD, 459 kcal/100 g, 20% protein, 40% carbohydrate, 40% fat; fat, D12 327, Research Diets, Inc., New Brunswick, NJ, USA) and normal water for 6 weeks (control group n=6). The amount of food was measured twice a week. The remaining six OVX mice were fed the HFHSD and soy isoflavone water (0.1%) (isoflavone group n=6) from 8 weeks of age for 6 weeks. The isoflavone group was given the same amount of food as the control group at the same age. The weight of both groups was measured twice a week.”

State the results of comparing food intake between the groups.

A limitation to add is that pair-feeding was not implemented. This would have ensured that outcomes were not affected by differences in food intake.

[Response]

Thank you for your excellent suggestion. We have added a note that there was no significant difference in food intake between the control and isoflavone groups.

We conducted pair feeding, so as you pointed out last time, we related the amount of food to body weight.

(3.Results Line 275-278)

“Since the mice were fed in pairs, no discernible differences were noted in the body weights of mice administered HFHSD with isoflavone (isoflavone mice) compared to those given HFHSD without isoflavone (control mice) (Figure 1a). In fact, there was no significant difference in food intake between the two groups (Figure 1b).”